# Reactivity of Covalent Fragments and Their Role in Fragment Based Drug Discovery

**DOI:** 10.3390/ph15111366

**Published:** 2022-11-08

**Authors:** Kirsten McAulay, Alan Bilsland, Marta Bon

**Affiliations:** 1Cancer Research Horizons—Therapeutic Innovation, Cancer Research UK Beatson Institute, Garscube Estate, Switchback Road, Glasgow G61 1BD, UK; 2Centre for Targeted Protein Degradation, University of Dundee, Nethergate, Dundee DD1 4HN, UK; 3Exscientia, The Schrödinger Building, Oxford Science Park, Oxford OX4 4GE, UK

**Keywords:** fragment-based drug discovery, fragment library, covalent fragments, electrophilic warhead, reactivity

## Abstract

Fragment based drug discovery has long been used for the identification of new ligands and interest in targeted covalent inhibitors has continued to grow in recent years, with high profile drugs such as osimertinib and sotorasib gaining FDA approval. It is therefore unsurprising that covalent fragment-based approaches have become popular and have recently led to the identification of novel targets and binding sites, as well as ligands for targets previously thought to be ‘undruggable’. Understanding the properties of such covalent fragments is important, and characterizing and/or predicting reactivity can be highly useful. This review aims to discuss the requirements for an electrophilic fragment library and the importance of differing warhead reactivity. Successful case studies from the world of drug discovery are then be examined.

## 1. The Role of Covalent Inhibitors and Warheads in Drug Design

The success of covalent drugs (Figure 1) has led to a renewed interest in the rational design of covalent inhibitors as therapeutic agents [1,2,3,4,5,6,7]. Amongst the multitude of drugs approved by the US FDA, over 40 form a covalent interaction with their target protein. Albeit, only a handful of these were designed to react in this manner [8]. In contrast to traditional reversible inhibitors, targeted covalent inhibitors (TCIs) are generally designed to form an irreversible bond with a specific amino acid upon binding to a protein target, effecting a biological response. This is made possible through the combination of a highly selective reversible motif with a reactive group by parallel optimization of both covalent and non-covalent binding interactions. Kinases represent one of the most highly targeted protein classes for covalent inhibition with a number of successfully approved FDA drugs [2]. Representative examples of rationally designed covalent inhibitors approved by the FDA are included in Figure 1.

Typically, nucleophilic amino acid residues are targeted [2], with cysteine being the most favored, due to its low abundance within the proteome and high reactivity [11,12,13]. A recently reported analysis suggests that 36% of covalent functional groups are designed to target cysteine, whilst far fewer are deployed against lysine (17%), serine (15%), tyrosine (8%), threonine (6%), histidine (5%), aspartate (4%), glutamate (4%), tryptophan (2%), methionine (1%), arginine (1%), and finally proline (1%) [14]. Utilization of a covalent mechanism of inhibition has the potential for increased potency and selectivity, with longer duration of action compared to non-covalent binders [15,16]. TCIs also offer an opportunity to overcome resistance mechanisms where binding site mutations have occurred, when the mutation does not affect the reactive amino acid or access to the binding site [17,18]. Despite these advantages, TCIs can show high toxicity due to off-target interactions [1,8,19]. Typically, these are associated with the inhibitor covalently binding to off-target proteins, causing an immunological response and/or cellular damage. Highly reactive intermediates may also form upon metabolism, with the potential for producing toxic effects. As such, PKPD and selectivity strategies to mitigate these risks have already been widely adopted within covalent inhibitor design [20]. The success and high affinities associated with TCIs has led to covalent PROTAC strategies being explored [21,22,23,24]. These bivalent molecules have the same advantages and disadvantages as standard TCIs but offer some added benefit over non-covalent PROTACs [25,26,27]. Lower molecular weight covalent PROTACs can be designed by exploiting the high affinity of the covalent fragment binding portion. Decreased weight and fewer H-bond donors can mitigate issues such as permeability, which are associated with the physicochemical properties of standard bivalent degraders. Due to their mode of action, covalent PROTACs also offer the ability to target proteins via ‘non-functional’ binding sites, increasing the number of potential targets across the proteome. Despite these advantages, addition of a covalent warhead negates the ability of the PROTAC to act catalytically, and so nullifies the general advantages of the modality. Namely, reduced potential dosage and off-target effects. Moreover, there have been conflicting reports as to the efficacy of the approach [28]. Reversible covalent PROTACs may offer the retention of catalytic function with the advantages of a covalent mode of action; recent studies have sought to explore this method [29,30,31].

To achieve covalent bond formation between a drug-like molecule and protein, it is often necessary to include a reactive ‘warhead’ within the structure. Several chemically distinct warheads have been utilized to this end. A selection of their structures and targeted amino acid residues are highlighted in Figure 2. Inherently reactive moieties such as epoxides and acrylamides have long been identified as covalent warheads. However, several new functional groups have recently been reported to capitalize on the nucleophilic nature of amino acids [32,33,34]. The majority of these approaches have reported novel cysteine reactive moieties [35,36,37,38,39,40,41]. Interestingly, the use of latent functionalities for bond capture have also been described. Mons et al. recently reported the use of an inherently unreactive alkyne moiety for covalent bond formation with Cys25 in Cathepsin K [42]. It should also be noted that not all warheads are designed to bind irreversibly. For example, aldehydes, boronic acids and cyanoacrylamides have been reported as reversible covalent binders [43,44,45,46,47,48]. Utilization of these warheads can allow optimization of binding parameters, including residence time within the protein, by altering the neighboring substituents and electronics of the warhead [46,49]. Despite the large selection of warheads reported in the literature, most rationally designed inhibitors which reach the clinic contain a Michael acceptor, largely due to their reasonable reactivity level, ease of synthesis and compatibility with DMPK.

Several mechanisms for covalent bond formation between amino acids and warheads have been identified and databases pertaining to these interactions have been developed. The CovPDB database is dedicated to high-resolution covalent protein-ligand complexes and has grouped reported reactions into 21 different mechanisms [50]. Comparatively, Keseru et al. recently developed the WHdb, a comprehensive list of warheads, for which they have defined 10 different reaction types [51]. Unsurprisingly, mechanisms which involve nucleophilic attack of the amino acid towards the warhead are the most abundant in both analyses. However, nucleophilic fragments targeting alternative bioactive molecules have recently been published. Matthews et al. have described the reaction of hydrazines with carbonyl groups in electrophilic co-factors for selective inhibition [52], whilst Ve et al. have demonstrated the reaction between an isoquinoline and NAD+ resulting in inhibition of SARM1 [53]. Despite hydrazines being commonly avoided by medicinal chemists, isoquinolines are present in numerous drug structures and so this may represent a potential future targeting strategy [54,55].

Some of the most abundant warheads (e.g., acrylamides, epoxides, sulfonyl fluorides) have been reported to form covalent bonds with a diverse range of amino acids. However, the recognition binding elements of the entire molecule are important and, as a result, rationally designed molecules are often highly selective for their targeted residue. Selectivity largely depends on the electronic properties, substituents, and steric effects, which in turn influence reactivity [56]. 60% and 55% of the warheads analysed in WHdb and CovPDB, respectively, were described as labelling only one amino acid [51]. A previous study by Klein et al. also showed that electrophilic fragments are perhaps less promiscuous than one might expect [57]. 72 fragment-like compounds, covering 6 different reactive electrophilic groups, were explored through a systematic study. 64% of the compounds examined were shown to bind to only one target residue. However, it should be noted that this does not necessarily reflect the overall landscape since commonly used warheads, such as acrylamide derivatives, can show less specificity [58].

## 2. Predicting Warhead Reactivity

Selection and optimization of TCIs is an open challenge in medicinal chemistry, because one should find a balance between binding affinity, specificity, inherent reactivity, and metabolic stability. Ideally, warhead reactivity should be reduced to a necessary minimum to prevent off-target reactions. In addition, the warhead reactivity range is influenced by the surrounding chemical functionality, i.e., by the electronic and steric effects of the scaffold. The reactivity of the partnered-amino acid will impact the reactivity level required from the warhead [59,60]. Particularly because amino acid nucleophilicity is greatly influenced by the chemical environment within the protein. The reaction mechanism can also affect the efficiency, especially in the case of multi-step reactions, where the role of intermediates must be considered [14]. For this reason, the ideal warhead should be adjustable, to meet the requirements of the target [32]. Finally, warheads should also be metabolically stable and non-toxic to allow in vivo use, as with any reversible inhibitor.

Several methods have been developed to determine and predict the relative reactivity of covalent warheads [58,61,62]. Experimentally, reactivity is usually measured using a kinetics-based assay with an amino acid surrogate. Both Nuclear Magnetic Resonance (NMR) and Liquid Chromatography Mass Spectrometry (LC-MS) can be used to monitor reaction over a given time to determine half-life through pseudo first-order kinetics [58,62,63]. In both cases, changes in peaks are monitored for disappearance of parent and appearance of product in the presence of an internal standard. Perhaps the most widely utilized method for cysteine reactive warheads is the measurement of the half-life of adduct formation with glutathione (GSH t½) [64,65,66,67,68]. This gives an idea of the relative reactivity of a warhead towards cysteine and acts as an indicator as to potential off-target reactivity and toxicology. However, reactivity does not always correlate with IC_50_ values for elaborated covalent inhibitors, as the non-covalent binding interactions largely determines selectivity and affinity [69,70,71]. Other experimental approaches have moved away from pseudo first-order kinetics. A photometric approach was previously developed to determine second-order reaction constants [64]. More recently, a high-throughput fluorescence-based thiol reactivity assay was developed to measure the reactivity of cysteine targeting fragments [72]. This method employs Ellman’s reagent (DTNB) and quantifies the kinetic rate constant based on the absorbance of its monomer TNB2- using second-order reaction rates. In 2018, Craven et al. reported a novel strategy for the optimization of covalent fragment kinetic selectivity ‘quantitative irreversible tethering’ (qIT) [73]. qIT uses a fluorogenic probe to determine a ‘rate of enhancement’ factor by comparing covalent bond formation between the fragment and both the target protein and GSH. Thereby, they accounted for the intrinsic reactivity of the warhead. A library of 138 covalent fragments was screened against CDK2 to demonstrate the method and a molecule showing five-fold rate enhancement for Cys177 in CDK2 over GSH was observed. In 2020, they reported an expansion of this work, including multiparameter kinetic analysis to determine the inhibition constant (k_i_) and inactivation rate constant (k_inact_) [74]. By merging two fragment scaffolds they were able to improve selectivity for the protein over GSH, thus demonstrating how the understanding of intrinsic reactivity is an important parameter in development. Experimentally measured quantities can be used to build Quantitative Structure–Activity Relationship (QSAR) models to help predict reactivity effects [75,76]. However, despite being commonly adopted, experimental data are subject to variable conditions, with compound stability and covalent bond reversibility being contributing factors [77]. To this end, several computational techniques have been explored.

As is normally the case in computational drug discovery, methods used to predict reactivity can be divided in to two main categories: Ligand-based and structure-based. Ligand-based methods require the definition of global reactivity descriptors. Ideally, these should be both simple and fast to determine ab-initio, via quantum mechanical (QM) calculations and should also be applicable to different classes of molecules. Both thermodynamics and kinetics play a role in covalent binding. Indeed, reaction energies and barriers have been computed from first principles to quantify the reactivity of Michael acceptors with methyl thiolate as a cysteine surrogate [78]. These parameters have been proven to correlate with GSH t_1/2_ for covalent fragments [79], thiol reactive inhibitors [61], acrylamides and 2-chloroacetamide warheads [77]. pKa’s of the amine precursors of acrylamides, as well as NMR shifts of the acrylamide alkene have also been demonstrated to be valuable descriptors for cysteine targeting warheads [79]. Proton affinity works well for a diverse range of small reactive fragments [46]. However, larger and more complex molecules cannot be described in this manner, because ligand conformational freedom must be taken into account [80]. Conformational sampling plays a role, especially when the calculation of the transition state is involved [77]. Based on this, it is beneficial to use truncation algorithms for drug-like molecules (>250 Da) [77,80]. Finally, the electrophilicity index correlates well with experiments, if calculated using only the warhead associated orbitals [81,82,83]. To date, none of the aforementioned descriptors have proven to be successful when used for a diverse library of molecules containing varied warheads. Moreover, it must be noted that most QM simulation protocols are carried out in the gas phase or implicit solvent. This approximation can fail when specific interactions between solute and solvent are important. To tackle these cases, one can adopt microsolvation models, nevertheless introducing an additional layer of complexity to the workflows [84]. Finally, although QM calculations are generally faster than experiments, they still require computational time and assessment of the suitable level of theory through benchmarking with experimental data. To overcome potential speed limitations, machine learning methods can be employed, where QM calculations are used to generate in silico training sets [77,85]. Artificial intelligence (AI) has recently attracted considerable attention in molecular design [86]. Although most applications to date have focused on non-covalent molecules, examples of AI in covalent inhibitor design have begun to emerge. Machine learning techniques can be used to combine different descriptors. For example, Palazzesi et al. [65] trained a random forest regression model to predict ab-initio calculated activation energies for acrylamides and 2-chloroacetamides using Dragon molecular descriptors. [http://www.talete.mi.it/products/dragon_description.htm; accessed on 30 August 2022].

An intrinsic limitation of ligand-based methods is represented (by definition) by the absence of the receptor. As previously stated, the reaction partner (the targeted amino acid together with its environment in the binding pocket) and mechanism can significantly affect covalent bond formation [87]. Structure-based approaches can address this aspect. However, the reactive nature of covalent bond formation cannot be properly captured by classical forcefields and requires explicit treatment of the electrons. The whole complex is too large to be fully described at QM level. This calls for a hybrid quantum mechanics/molecular mechanics (QM/MM) approach where the targeted amino acid (and eventually a part of its surrounding), together with the inhibitor, are described using QM, whilst the rest of the system, such as the solvent and other residues of the protein, are described using an MM model [88,89,90]. QM/MM methods have been used to characterize the whole binding of the covalent adduct [88,91]. Additionally, these methods can be successfully coupled with Fragment Molecular Orbital (FMO) analysis to investigate the mechanism of reaction and to evaluate the interaction energy of ligands with single residues. This way, one can identify and characterize promising regions where additional binding energy can potentially be gained [92,93]. QM/MM calculations can also be combined with enhanced sampling techniques, to obtain binding free energies and rate constants [91,94,95]. Although very powerful, QM/MM calculations are computationally intensive and require a careful, non-straightforward initial set-up. As a result, they are unsuitable for virtual screening purposes. Covalent and reactive docking algorithms have instead been used with some success [88,96,97,98,99,100,101,102,103,104,105,106]. However, they have not yet been validated over a large class of receptors and ligands. Moreover, they present similar limitations to the non-covalent docking methods, such as limited force field accuracy and the lack of a proper description of a receptor’s flexibility [107]. Whilst developers are continuously improving the algorithms, users customize the methods for challenging systems by developing new computational strategies that include molecular dynamics and MM/GBSA calculations [108]. Docking calculations can also be used for the generation of docking-based pharmacophores in QSAR models [109].

Despite significant advances in the field, the overall complexity of covalent binding means there is still a lot of work to be done. Currently, there is not a straightforward combined experimental and computational solution to accurately measure and predict the binding of different warheads with different targets and amino acid residues. However, the methods described are incredibly useful and can help guide fragment optimization.

## 3. FBDD and the Role of Covalents within

Fragment-based drug discovery (FBDD) is a highly successful and complementary method to high-throughput screening (HTS) for the discovery of bioactive molecules for a drug discovery campaign [110,111]. It has been widely adopted in both academic and industrial institutions [112,113] due to its ability to sample a vast amount of chemical space with a ‘small’ number of compounds. Fragment-like molecules are more likely to bind in an atom-efficient manner [114,115]. Consequently, a library of one to two thousand compounds can easily provide quality hit matter [116]. Therefore, good library design, incorporating a diverse range of pharmacophores with synthetically accessible growth vectors, is highly important to identify high quality leads [107,117]. There have been six FDA approved drugs to date which were discovered through an FBDD approach. Perhaps the most notable is sotorasib which irreversibly binds to KRAS_G12C_, a target previously considered undruggable [118].

A fragment is typically defined as a small molecule with ≤20 heavy atoms and MW ≤300 Da. Fragment physicochemical properties are important to ensure efficient screening and hit-to-lead campaigns and so the ‘rule of three’ (Ro3) was devised to guide library design [119]. This states that a molecule should have ≤3 hydrogen bond donors (HBD), ≤3 hydrogen bond acceptors (HBA) and a computed logarithm of the partition or distribution coefficient (cLogP/cLogD) of ≤3. Additional criteria include ≤3 freely rotatable bonds and a polar surface area (PSA) of ≤60. However, these criteria have progressed, and efficacious fragment hits often violate at least one of these rules [120,121].

Fragments hold particular utility in the identification of ‘hidden’ binding pockets, e.g., allosteric sites or ‘hot spots’ implicated in protein–protein interactions [122]. Furthermore, hit rates can be an indication of the overall ligandability of a target [123]. Hits from HTS campaigns generally display dissociation constant values (K_d_) in the nM-µM range, whilst reversible fragment hits tend to have weak affinities, typically µM–mM. This often means more extensive chemistry efforts are required to generate a lead-like molecule. The weak affinities observed for reversible fragments require biophysical techniques such as surface plasmon resonance (SPR), NMR, X-ray crystallography and thermal shift assays are typically to measure binding. It is best practice to use two orthogonal methods to validate hits. Biochemical assays, which are generally used for HTS screens, are not typically sensitive enough to detect K_d_ values within the fragment range.

TCIs are often obtained upon modification of reversible ligands (‘binder-first’ approaches), by attaching a warhead to improve target selectivity and efficiency. ‘Binder-first’ approaches have two intrinsic limitations: (1) the necessary existence of a non-covalent binder as a starting point and thus, applicability only to ‘traditional’ binding sites that can already host non-covalent ligands; and (2) that those ligands must be close to a reactive amino acid residue. Covalent FBDD can overcome these two limitations and was recently used on protein targets that lack well-defined binding pockets, often classified as ‘undruggable’ [118]. It has also proven particularly useful for screening beyond substrate pockets, the so-called ‘cryptic pockets’ [124,125], and has been utilized to improve enzyme activity [126]. Although it is out with the scope of this review, it is worth noting that covalent fragments have also opened the door to novel target identification through chemoproteomics. Instead of screening against one target, as in traditional FBDD, covalent fragments have been used to identify potentially druggable proteins within the proteome [33,127,128,129,130,131,132,133,134,135].

As well as considering the standard FBDD library design criteria, further thought must be given to the design of covalent libraries. The reactive nature of covalent fragments means that stability is often a concern, both inherently and under physiological conditions. For example, Grygorenko et al. designed a library of 62 sulfonyl fluorides for screening against trypsin, but noted limited stability of some of the fragments in DMSO [136]. Moreover, the reactivity, size and functionality of the electrophilic group should be taken into consideration. Desirable parameters may change depending on the targeted amino acid [137] and its position within the protein [138]. Amino acid nucleophilicity can vary substantially, depending on the protein environment [59,63]. Thus, amino acids with a higher pKa may require a more reactive warhead for efficient reaction. In addition, the positioning of the electrophilic group within the fragment should be considered. The warhead geometry and angle of attack of the amino acid significantly affect covalent bond formation and so attachment via a minimal linker allows easier access to the warhead than when embedded within the scaffold. Shokat et al. reported changes in the ligand-binding mode and labelling of KRAS_G12C_ with different warhead chemotypes [139]. This study demonstrated the utility of using a diverse range of warheads and highlighted the significance of obtaining optimal geometry between warhead and nucleophile. A library which includes fragments with a range of reactivities [79] and electrophilic groups [61] is therefore beneficial since there is no ‘one size fits all’ warhead. Several covalent fragment libraries are now commercially available, with size, diversity and electrophilic warheads differing in each [140]. However, designing a bespoke set with a high level of diversity to fit screening criteria is often most beneficial [107].

Screening of very highly reactive fragments may lead to lower affinity ‘reversible’ binders, with k_inact_ having a more substantial role in the overall binding recognition. In particular, multiple labelling can represent absence of specificity. However, promiscuity for moderately active electrophiles may have previously been overestimated [57]. London et al. recently screened a library of 993 mildly electrophilic fragments against 10 protein targets with a hit rate comparable to normal reversible FBDD screens [72]. Elaboration of fragments led to the identification of potent selective probes for two of the enzymes which previously had no known inhibitors. This work highlighted that the reactivity of each fragment does not necessarily correlate with its promiscuity.

Hits identified from a covalent fragment screen can be grown and merged using conventional fragment strategies [58]. Nevertheless, analysis of covalent fragment hits should be thorough to ensure binding occurs within a ‘real’ site and is not a result of high fragment electrophilicity. In general, covalent fragments have higher affinity and selectivity compared to non-covalent FBDD hits due to the irreversible nature of the binding. Furthermore, contrary to non-covalent FBDD, the binding motifs might not significantly change upon growing due to retention of the warhead. Improving the non-covalent binding affinity, K_i_, can allow for the removal of the warhead upon optimization, with potency retention. However, pursuing a TCI approach is increasingly popular [141]. A covalent approach may therefore be favored to identify hits for lower affinity allosteric sites. As previously discussed, this strategy is only suitable if a reactive nucleophilic residue is present and there is therefore a natural bias as to what can be targeted in this way. Efforts have been made to overcome this using photoaffinity labelling, whereby photoreactive fragments, upon irradiation with light, crosslink to proximal protein residues. Cravatt et al. initially reported fragment-protein interactions in live cells [142] and more recently, Bush and co-workers have utilized ‘Phabits’ to enable high-throughput screening against purified POIs. Hits were identified by intact protein LC-MS, with follow-up studies to ascertain binding affinity and the site of crosslinking. Photoactivated covalent capture of DNA-encoded fragments for hit discovery has also been described by Ma et al. [143] Utilizing diazirine moieties the group identified fragments for PAK4 and the bacterial enzyme 2-epimerase which were validated as hits by NMR and crystallography. Despite potential future utility, photoreactive fragments are currently widely non-commercial and crosslinking yields can be low and do not always correlate with affinity [144].

Historically, covalent fragment screening was largely limited to disulfide tethering methods and was used for proteins with both native and engineered cysteines [145,146,147]. In 2013, Shokat et al. even utilized this method to identify binders of KRAS_G12C_, demonstrating its druggability [148] and paving the way forwards for the discovery of sotorasib and other small molecule inhibitors [118]. However, although this example demonstrates the method’s utility, it also highlights its pitfalls. Following the identification of a disulfide hit, a library of fully irreversible electrophiles was needed to progress the project and identify an inhibitor. As such, modern screening methods generally utilize irreversible electrophilic warheads directly. In principle, fragments that can irreversibly bind to their target can overcome the low affinity that limits non-covalent fragment screening and can be screened at lower concentrations. This therefore makes them amenable to alternative screening methods beyond the conventional biophysical assays. For example, KRAS_G12C_ binders were identified using a nucleotide exchange assay through Carmot Therapeutics ‘Chemotype Evolution’ technology which generated a library of ‘beyond rule of 3′ fragments by pharmacophore linking [149]. Traditional techniques are also used. NMR screening is often easier for covalent fragments due to increased chemical shift perturbation allowing for easier analysis [150]. Several high-profile targets have been screened in this manner, e.g., bromodomain containing protein 4 (BRD4) [125,151]. Nonetheless, high-throughput screening is most often carried out through liquid chromatography with tandem mass spectrometry (LC-MS/MS) to determine covalent binding [148]. Native MS can be combined with time-of-flight (TOF) instruments, thus increasing detection of both the target and fragments [152]. ‘Cocktails’ of fragments with different masses can be screened to increase throughput and the exact amino acid which is labelled can be determined by a digestion protocol. Moderate to large sized fragment libraries have been screened in this manner to identify hits for well-known targets such as Janus Kinase 3 (JAK3) and KRAS [148,153].

In silico screening is also becoming an ever-expanding tool for covalent fragments [100,154]. CovaDOTS and Cov_FB3D are among the computational frameworks developed specifically for in silico covalent FBDD [155,156]. CovaDOTS uses available fragment hit information to create covalent analogues, utilizing structure-based molecular modelling and chemistry knowledge. It consists of two stages. The first, the growing stage, generates a library of compounds using common synthetic routes from an active fragment and a source of available building blocks. The second, the linking stage, covalently attaches the library to a given nucleophilic protein residue through virtual screening, where the protein residue is treated as the second ‘fragment’. Comparatively, Cov_FB3D involves de novo in silico assembly of covalent inhibitors and consists of three main stages. At first, a library of warhead fragments is covalently docked to a receptor. Secondly, a library of non-covalent fragments is docked, and a non-covalent substructure is generated by in silico assembly of the fragments displaying the highest scores. Finally, the covalent fragment and non-covalent substructures are separately scored. The best non-covalent poses are linked to a selection of covalent fragments to generate possible covalent inhibitors, for which a synthetic accessibility measure is computed. To the best of our knowledge, both methods have only been assessed retrospectively, hence their capability in determining new covalent hits have still to be tested. Fragment-based design was also recently combined with deep generative AI to design new covalent BTK inhibitors using a “deepSARM” (SAR-matrix) approach [157]. The method utilizes a 2-step hierarchical decomposition of compounds into cores and substituents. In the first step, matched molecular series are generated, differing only by the substituent at a single site for each unique core. In the second step, fragmentation is repeated on the cores obtained in the first step. The generative model recombines the second-round core and substituents to yield a first-round core. The generated second round substituent is required to contain the warhead of interest. First-round cores are further decorated with additional substituents, increasing diversity. The model was initially trained on a large kinase inhibitor dataset and subsequently fine-tuned using known covalent BTK inhibitors. The candidate molecule set was further refined based on pharmacophore models of ibrutinib bound to BTK.

## 4. Covalent FBDD Case Studies

There have been several successful covalent FBDD campaigns to date, many of which demonstrate the utility of covalent FBDD to generate hits for difficult targets [140,152]. A variety of warheads with a breadth of reactivities have been demonstrated to bind to numerous protein targets. This section aims to summarize some of the most recent and successful examples.

### 4.1. GTPases

Arguably the most successful covalent fragment story is that of the KRAS_G12C_ inhibitors. KRAS is one of the most frequently mutated oncogenes, playing a role in numerous highly fatal cancers [158]. In 2013, Shokat et al. described the use of disulfide tethering to identify binders of KRAS_G12C_ [148]. Replacement of the thiol in an initial hit with an acrylamide, and a few other functional group changes, led to a molecule which displayed some cellular potency. Optimization of this scaffold ultimately led to the discovery of ARS-1620, described by Liu et al. in 2018 [159]. Separately, a collaboration between Carmot therapeutics and Amgen utilized Carmot’s ‘Chemotype Evolution technology’ to identify novel KRAS_G12C_ binders. This methodology entailed rapid synthesis and testing of beyond rule of 3 fragment-like molecule libraries based on pharmacophore linking [149]. Screening of the unpurified acrylamide compounds and subsequent optimization led to the discovery of **1** (Figure 3) which was highly potent but had poor bioavailability. Ultimately, they were able to learn from the ARS-1620 binding mode and grow into the so called ‘cryptic pocket’, resulting in the discovery of AMG-510 [118]. Notably, it took only 8 years from the initial publication by Shokat, demonstrating covalent binding to G12C, to sotorasib (AMG-510) gaining FDA approval for treatment of non-small cell lung cancer (NSCLC) [160,161].

Further research has sought to exploit alternative covalent strategies to target other GTPases implicated in oncology. Meroueh et al. recently reported covalent fragment screening to identify binders of Rgl2 to inhibit Ral GTPase activation [162]. Ral GTPases belong to the RAS superfamily and are directly activated by KRAS. Several chloroacetamide and acrylamide fragments were identified to bind allosterically at Cys284, inhibiting Ral GTPase exchange. Although Cys284 is not located at the Ral-Rgl2 interface, it is part of a loop where several residues encounter the GTPases. Indoline fragments proved most potent, with EC_50_s in the micromolar range and could provide a starting point for fragment expansion to provide more potent and selective inhibitors.

Tate and co-workers identified the first structurally validated covalent ligands of Rab27A, a small GTPase which promotes growth and invasion of numerous cancer types, using covalent fragment screening [163]. The nature of the Rab27A-effector PPI interface coupled with its high affinity for GTP make it a highly challenging target for traditional hit finding strategies. By utilizing a covalent fragment approach and qIT, the team were able to identify binders of both Cys188 and Cys123 in the SF4 pocket which are unique to Rab27A and 27B within the sub-family. The authors acknowledge PAINS and cytotoxicity issues for the reported binders, nonetheless, this result further highlights the utility of covalent FBDD to identify binders for challenging targets and structural data may provide a strategy for future molecule elaboration.

### 4.2. SARS-Cov-2

COVID-19, caused by the novel coronavirus SARS-Cov-2, resulted in thousands of deaths and millions of infections in early 2020, causing a global pandemic [164]. At the time there were no known treatments or vaccines, and so SARS-Cov-2 main protease became a very high-profile target. Several groups sought to use high-throughput and computational screening strategies to try and identify small molecule inhibitors of SARS-Cov-2 proteases [165,166,167,168,169], with many looking to exploit a covalent strategy [170,171,172,173,174,175,176]. Covalent fragment screening against SARS-Cov-2 main protease was initiated shortly into the pandemic by a group of academics from numerous institutions [177]. Several other cysteine proteases have previously been targeted using an electrophilic fragment strategy [72,178,179,180,181]. The screen was carried out by X-ray crystallography, in collaboration with Diamond Light Source, initially using the electrophilic fragment library described by Resnick et al. [72]. Covalent hits were then screened alongside a set of 1176 non-covalent compounds compiled from several libraries. The team found 48 covalent hits in the active site, as well as a further 23 non-covalent binders. The covalent hit-rate equated to 8.5% with *N*-chloroacetylaniline and *N*-chloroacetyl-*N*’-sulfonamido-piperazines proving to be frequent hitters. Interestingly, non-conventional covalent hits were identified in the screen **3**–**6** (Figure 3). Most notably, 3-bromoprop-2-yn-1-yl amides of N-acylamino acids **5** and **6** from the PepLites collection [182], which have inherently low intrinsic reactivity [42], bind to Cys145 in the active site with elimination of bromide (Figure 4b). A fragment merging strategy may be possible from the structural information collected and could contribute towards a SARS-Cov-2 treatment in the future.

### 4.3. BRD4-BET2

Bromo- and extra-terminal (BET) domain oncogenic networks are activators in several cancer types. As a result, BET-bromodomain inhibitors have become an increasingly promising class of anti-cancer agents [183]. Despite this, selectivity can often be an issue, with toxicity reported in clinical trials [184]. In 2020, Smith et al. reported an electrophilic fragment screen of 200 acrylate methyl esters to identify selective binders of BRD4-BD2 [125]. The screen was carried out using mass spectrometry and identified 7 initial hits with selectivity towards BRD4-BD2 over the closely related BRD4-BD1 and BRD3-BD2, including acrylate methyl ester **7** (Figure 5). Subsequent MS/MS fragmentation and NMR studies identified Cys356 as the labelled residue. Cys356 is unique to BRD4 [185] and sits adjacent to the acetyl-lysine binding site, thus, a linking strategy with a known BRD4 inhibitor (**JQ1**) [186] was followed. The resultant **8** showed cellular activity, targeting both BD1 (via JQ1) and BD2 (via the covalent), and represents the first chemical probe capable of binding orthogonally to the acetyl-lysine site.

### 4.4. Enzymes Involved in Ubiquitination/Deubiquitination

With the rise of protein degradation strategies and a deeper understanding of the role of deubiquitinating enzymes, targeting of E3 ligases and DUBs has become increasingly popular, with a few covalent fragment strategies now published [128,187,188]. Resnick et al. used their high-throughput fluorescence-based thiol reactivity assay, previously discussed, to identify ligands for DUBs OTUB2 and USP8 [72]. Screening against OTUB2 generated 47 hits with binding >50%, whilst 20 hits were found for USP8; however, 13 out of the 20 hits were found to be promiscuous. In contrast, 42 of the OTUB2 hits were found to be non-promiscuous. High-throughput crystallographic information allowed a selective probe to be quickly generated when no previously known inhibitors had been reported. Additionally, in 2019, Johansson et al. described a fragment based covalent ligand screen for the discovery of RBR E3 ligase [189]. A library of 106 compounds containing α,β-unsaturated methacrylate motifs was generated, based on the GSK fragment collection. Fragments were primarily screened as cocktails using intact protein LC-MS. Protein crystallography showed binding to the active site cysteine of the catalytic HOIP subunit. This information could hold potential for future structure-based development of covalent inhibitors for RBR E3 ligase.

### 4.5. Pin1

Recently, Dubiella and co-workers described a covalent fragment screen for Pin1, a phosphorylation dependent proline isomerase implicated in oncogenic processes [190]. Its shallow binding pocket is positively charged making it a traditionally difficult target for medicinal chemists. The group screened their fragment library, described in previous work [72], via intact protein LC-MS and identified 111 binders, with 48 hits exhibiting >75% protein labelling. Sulfolane containing chloroacetamides were found to be frequent hitters. Thus, close analogues were investigated, ultimately resulting in the discovery of Sulfopin (Figure 6a). Sulfopin displays reduced intrinsic thiol reactivity whilst maintaining potency and selectivity. Selectivity for Pin1 was demonstrated using chemoprotoeomics and no general toxicity was observed. The group was also able to demonstrate in vivo target engagement and tumor reduction in a mouse model, highlighting the utility of covalent fragment screening to identify in vivo tool compounds.

### 4.6. GPX4

In 2022, Cordon et al. ran a phenotypic covalent fragment screen to identify molecules which differentially affected HepG2 liver cells under hypoxia and normoxia. 930 electrophilic fragments, encompassing 6 different warheads, were analyzed, with 49 displaying the desired phenotype (5% hit rate) [191]. These 49 were further investigated at lower concentrations, with an additional sub-set retested in 8-point serial dilution, resulting in 2 confirmed normoxia-selective fragments. As was seen with other screens discussed, a series of molecules containing a specific motif were identified; in this case, propiolamides. Propiolamide **9** showed semblance to a known GPX4 inhibitor with an alternative masked nitrile-oxide electrophilic warhead (**ML210**) and thus was investigated as the primary target [192]. CETSA, fluorescent labelling and Western blot experiments validated GPX4 as the target for the phenotypic fragment hits, with induction of ferroptosis in cells also observed. More recently, 2-alkynylthiazoles have also been reported as novel warheads for this target [193]. 

### 4.7. LP-Pla2

Unlike the other studies discussed in this section targeting cysteine, Huang et al. utilized a covalent fragment approach to target a catalytic serine, i.e., S273 in the binding pocket of the serine lipase Lp-PLA2 [194]. Instead of running a screen, the group chose to focus on a single novel serine reactive warhead, an enol cyclocarbamate fragment **10**, derived from the natural product DSM-11579 (Figure 7). Fragment growing and merging with a known reversible fragment **11**, part of the darapladib scaffold, gave amine **12**. Further optimization, guided by structural information, resulted in a trifluoroethyl ether **13** which displayed 130,000-fold and 39,000-fold increased inhibitory activity and selectivity, respectively, over PLA2VIIB, a homologous protein. Trifluoroethyl ether **13** was also found to have selectivity over a wide range of other serine hydrolases, making it an ideal candidate for future exploration.

### 4.8. Tau

Traditional medicinal chemistry strategies are generally less efficacious against disordered proteins due to the lack of distinct 3D structures. However, covalent fragments have the propensity to overcome this. Petri et al. have reported a covalent strategy to target the intrinsically disordered protein Tau, an endogenous protein present in the CNS, indicated in neurodegenerative diseases [195]. Initially, the group mapped the available cysteines and prioritized a library of warheads using orthogonal biochemical and biophysical methods (intact protein MS, ^19^F NMR and a fluorescence based Ellman’s assay). The library contained 25 warheads, described in previous publications [196,197], and resulted in initial prioritization of 3 warheads. A vinylsulfone was ultimately selected due to its superior stability, with labelling of both Tau-K18 cysteines (C291S and C322S) confirmed by 2D NMR experiments. Subsequent addition of the vinylsulfone warhead to known, and predicted, reversible binders of Tau resulted in covalent inhibitors which induced conformational changes in Tau and reduced aggregation. The strategy may provide a way forward for other intrinsically disordered proteins and serve as a starting point for developing Tau therapeutics.

## 5. Conclusions

Covalent FBDD has proven itself highly useful for the identification of novel binders for a multitude of proteins. Recent case studies have found several structurally diverse warheads as potent hits, highlighting the need for libraries to contain a diverse range of electrophilic moieties with a range of reactivities which can cater for differing amino acid nucleophilicities within protein environments. Inherent reactivity does not have to be high to generate valuable hits, as demonstrated by the fragment Sars-Cov2 binders. Likewise, highly reactive warheads have demonstrated relatively low levels of promiscuity in screens. Regardless, understanding reactivity is highly beneficial when driving a program forwards. Striking a balance between reactivity, potency and selectivity is key to identifying potential candidates and minimizing off-target effects. To this end, several methods have been developed and discussed. However, work is still needed to identify more generalized protocols. Nonetheless, we believe covalent FBDD will continue to grow as a valuable tool for hit identification and the unique nature of covalent ligands will drive generation of hits for currently undrugged and complex protein targets. In particular, we look forward to seeing further exploration of alternative warheads, both reversible and irreversible, which may have the potential to optimize the balance between potency and selectivity. In general, we hope that the successes discussed within this review will continue to inspire further evolution of covalent fragment-based approaches, for both traditional FBDD and newer technologies such as chemoproteomics and foresee its advances.

## Figures and Tables

**Figure 1 pharmaceuticals-15-01366-f001:**
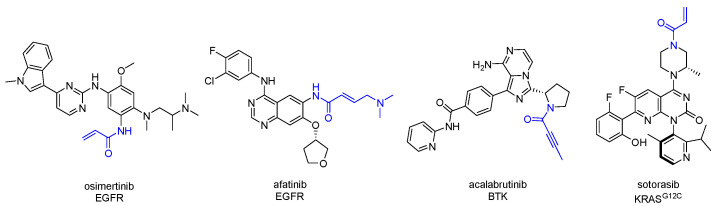
Representative examples of FDA approved covalent inhibitors Osimertinib (2015), afatinib (2013), acalabrutinib (2017), and sotorasib (2021) alongside their targeted proteins. Reactive groups are highlighted in blue [2,9,10].

**Figure 2 pharmaceuticals-15-01366-f002:**
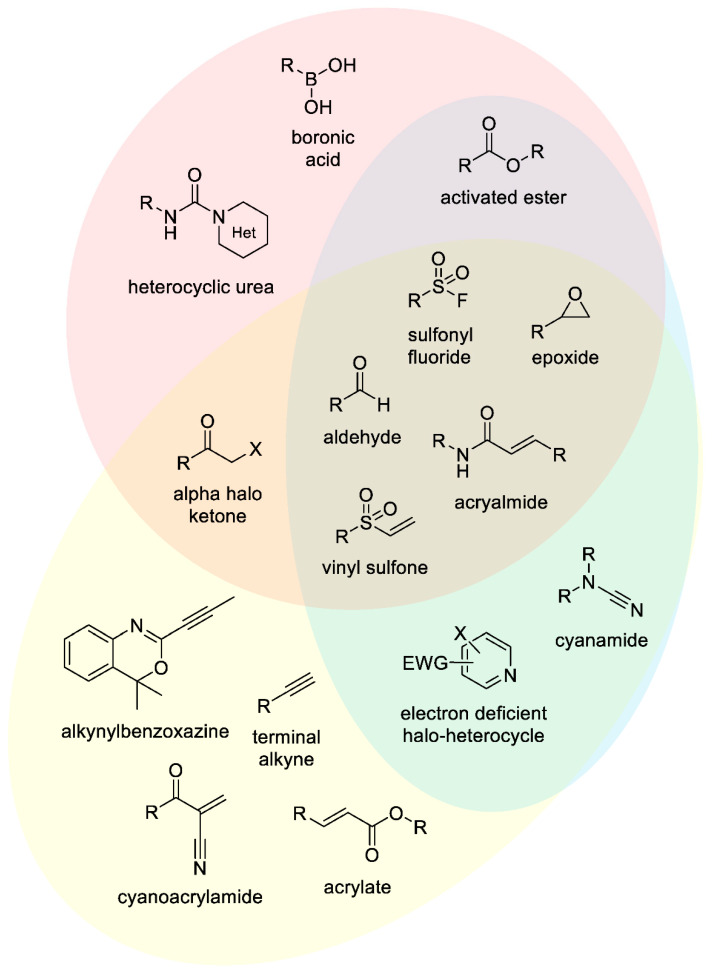
Representative warhead motifs and overlap of amino acid activity. Colours represent selected amino acids which have been labelled in the literature by those warheads; yellow—cysteine, blue—lysine, red—serine/threonine. Where overlap occurs the corresponding secondary colour is observed [14,36].

**Figure 3 pharmaceuticals-15-01366-f003:**
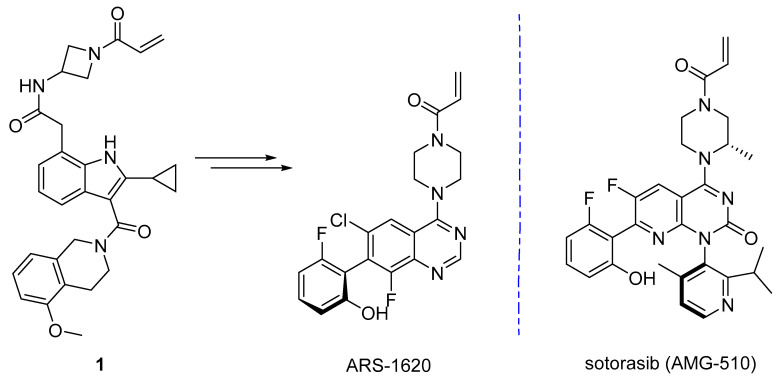
KRAS_G12C_ covalent inhibitors.

**Figure 4 pharmaceuticals-15-01366-f004:**
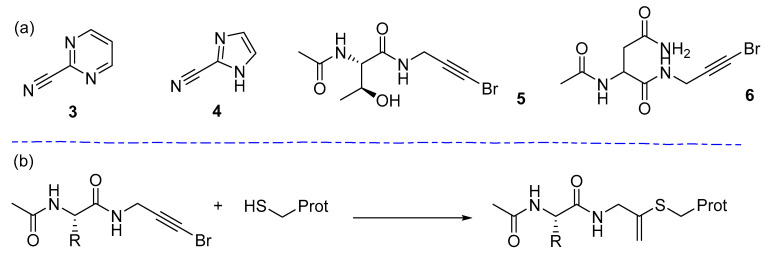
(**a**) Covalent fragment binders of Sars-Cov2 main protease; (**b**) Reaction scheme of covalent modification to Cys145 by 3-bromoprop-2-yn-1-yl amides.

**Figure 5 pharmaceuticals-15-01366-f005:**
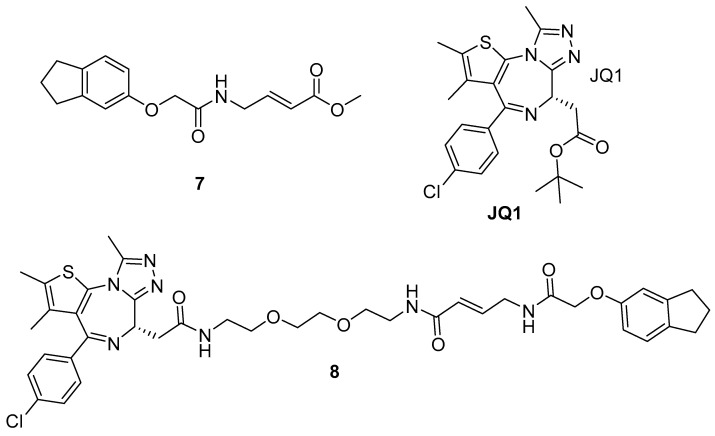
Small molecule binders of BRD4.

**Figure 6 pharmaceuticals-15-01366-f006:**
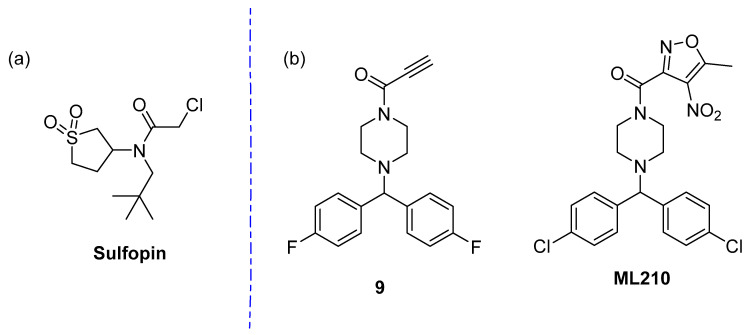
(**a**) Structure of sulfopin; (**b**) Structures of GPX4 covalent binders.

**Figure 7 pharmaceuticals-15-01366-f007:**
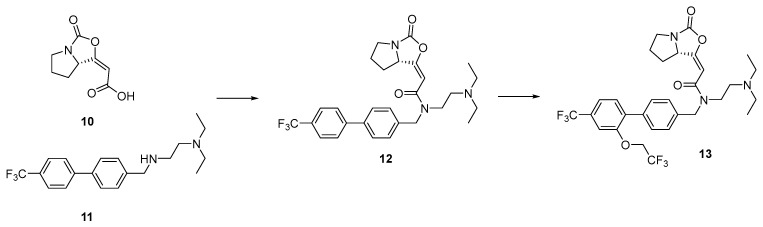
Fragment merging and elaboration of LP-Pla2 small molecule inhibitors.

## Data Availability

Data sharing not applicable.

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
