# Peer review of "Reactivity of Covalent Fragments and Their Role in Fragment Based Drug Discovery"

_pharmaceuticals, 2022, doi:10.3390/ph15111366_

Round 1

Reviewer 1 Report

This paper deals with a "Reactivity of Covalent Fragments and Their Role in Fragment Based Drug Discovery". I have no issues at all with the methods, data analysis, or conclusion.

1-The writing can be polished and any grammar errors should be fixed completely.

2-The novelty should be emphasized because it is not obvious.

Author Response

Thank you very much for reviewing our paper. We have further proof read the document and sought to rectify any grammatical errors as suggested. 

We appreciate the comment regarding the novelty of the article. Although other review articles have been published in the field, we believe that we have given a unique and concise overview of covalent fragment reactivity and recent FBDD efforts. In our opinion, the papers discussed within the review emphasize the strength of the field and the novel studies which have been conducted in recent years.  

Reviewer 2 Report

This manuscript is interesting and could be accepted to Pharmaceuticals. The topic is definitely original and actual. Fragment based drug discovery has long been used for the identification of new ligands and interest in targeted covalent inhibitors has continued to grow in recent years; with high profile drugs such as osimertinib and sotorasib gaining FDA approval. It is therefore unsurprising that covalent fragment-based approaches have become popular and have recently led to the identification of novel targets and binding sites, as well as ligands for targets previously thought to be ‘undruggable’. Understanding the properties of such covalent fragments is important, and, in particular, characterizing and/or predicting reactivity can be highly useful. This review aims to discuss the requirements for an electrophilic fragment library and the importance of differing warhead reactivity. Successful case studies from the world of drug discovery are then be examined. 

The introduction provide sufficient background. The research methodology is adequate and modern. The results are clearly presented. The amount of data is large. The conclusions supported by the data. The manuscript good illustrated and interesting to read. English language and style are fine, and may be very minor polishing from native speaker is recommended. I have also couple of minor suggestions:

- Following relevant article could be cited to show other approaches to theoretical prediction of electrophility/nucleophility: J. Organomet. Chem. 2015, V. 797. P. 8.

- Some perspectives regarding the future research could be formulated in conclusions section.

Overall, this nice manuscript could be accepted for publication after minor revisions.

Author Response

Thank you very much for reviewing our paper and for the positive comments. We have further proof read the document and sought to polish the language as suggested. Also, thank you very much for highlighting the interesting J. Organomet. Chem. Paper. In this instance, we felt that the paper fell slightly outwith the scope of our discussion and have therefore not included it as a reference. 

We have added a few sentences at the end of the conclusion to detail our perspectives regarding future advances in the field. We hope this is of interest and meets your expectations.  

Reviewer 3 Report

As such, the manuscript is of highest interest and quality so that it should be published as it is.  The manuscript is sound and well written

There are two minor comments, both of which I would really like to see be taken into consideration.

1. Covalent proteolysis targeting chimeras combine the cutting edge research areas of targeted covalent inhibitors (TCIs) and proteolysis targeting chimeras (PROTACs). This nascent field of research should also be addressed in this Review by the authors and the potential advantages and disadvantages of this approach should be discussed.

2. The authors are kindly requested to overall cross-validate the Bibliography.  For instance, Ref. n° 168 is a duplicate of Ref. n° 32.

The authors may wish to include the following References:

Ispinesib as an Effective Warhead for the Design of Autophagosome-Tethering Chimeras: Discovery of Potent Degraders of Nicotinamide Phosphoribosyltransferase (NAMPT). DOI: 10.1021/acs.jmedchem.1c02001

Ynamide Electrophile for the Profiling of Ligandable Carboxyl Residues in Live Cells and the Development of New Covalent Inhibitors.                               DOI: 10.1021/acs.jmedchem.2c00272 

Chemical proteomic identification of functional cysteines with atypical electrophile reactivities. DOI: 10.1016/j.tetlet.2021.152861

Improved Electrophile Design for Exquisite Covalent Molecule Selectivity. DOI:10.1021/acschembio.1c00980

Assessment of Reversibility for Covalent Cysteine Protease Inhibitors Using Quantum Mechanics/Molecular Mechanics Free Energy Surfaces.                DOI: 10.1021/acs.jcim.2c00466

Lysine-Targeted Reversible Covalent Ligand Discovery for Proteins via Phage Display. DOI: 10.1021/jacs.2c07375

A Warhead Substitution Study on the Coronavirus Main Protease Inhibitor Nirmatrelvir.  DOI: 10.1021/acsmedchemlett.2c00260

Expedited Approach toward the Rational Design of Noncovalent SARS-CoV-2 Main Protease Inhibitors. DOI: 10.1021/acs.jmedchem.1c00509

3CL Protease Inhibitors with an Electrophilic Arylketone Moiety as Anti-SARS-CoV-2 Agents. DOI: 10.1021/acs.jmedchem.1c00665

Discovery of Chlorofluoroacetamide-Based Covalent Inhibitors for Severe Acute Respiratory Syndrome Coronavirus 2 3CL Protease.                                      DOI: 10.1021/acs.jmedchem.2c01081

Targeting SARS-CoV-2 Main Protease for Treatment of COVID-19: Covalent Inhibitors Structure–Activity Relationship Insights and Evolution Perspectives. DOI:10.1021/acs.jmedchem.2c01005

Author Response

Thank you very much for reviewing our paper and for the positive comments. Also, thank you for the list of relevant references. We have included these within the manuscript where appropriate and have also resolved the duplication issues within the bibliography. Thanks very much for bringing this to our attention.  

We agree that covalent PROTACs combine two cutting edge research areas and is a topic very worthy of discussion. However, we did not originally include this area as we had hoped to focus more on the small molecule/fragment aspect. We have now included a small section within our introduction detailing some of the advantages/disadvantages of a covalent PROTAC approach. We hope this gives a brief but informative overview of the field.  

Reviewer 4 Report

This minireview summarizes the recent developments achieved with electrophilic fragments starting from the rationale of combining the benefits of covalent mechanism of action and fragment-based drug discovery. After giving an overview on both the theoretical and experimental approaches to estimate warhead reactivity the authors analysed a number of covalent FBDD case studies that is clearly the most useful part of the review.  These studies show that the approach might be useful for challenging targets including GTPases, SARS-Cov-2 enzymes, ubiquitinating and deubiquitinating enzymes and intrinsically disordered proteins. Furthermore, it can be applied in both target based and phenotypic screening targeting Cys residues and beyond. Altogether, the manuscript gives a fair and well-balanced overview of a dynamically developing field with already important applications in both medicinal chemistry and chemical biology. The only suggestion is to update some certain references such as No. 7, 24, 95, 131, and 179. The manuscript is generally well written and should be published after this minor revision.

Author Response

Thank you very much for reviewing our paper and for the positive comments. We have updated the references which you highlighted within the bibliography where possible. Unfortunately, not all of the recent articles have page and volume numbers yet. In these cases, we have now included the DOI within the reference.  

Reviewer 5 Report

In this work, the author highlighted on the case studies from the world of drug discovery, explored the reactivity of covalent fragments and their role in drug discovery. The author highlighted different factors that need to be considered for to understand it. The author also provided very nice scheme to highlight potential drugs and covalent fragments.

This is nicely done and potential for publication.

Author Response

Thank you very much for reviewing our paper and for the positive comments.